# The Other Side of the ACEs Pyramid: A Healing Framework for Indigenous Communities

**DOI:** 10.3390/ijerph20054108

**Published:** 2023-02-25

**Authors:** Maegan Rides At The Door, Sidney Shaw

**Affiliations:** 1National Native Children’s Trauma Center, University of Montana, Missoula, MT 59812, USA; 2School of Counseling, Walden University, Minneapolis, MN 55401, USA

**Keywords:** Indigenous Wellness Pyramid, ACEs, holistic, social ecological framework, historical trauma

## Abstract

For over two decades, extensive research has demonstrated significant associations between adverse childhood events (ACEs) and a wide range of negative health, mental health, and social outcomes. For Indigenous communities globally, colonization and historical trauma are commonly associated with ACEs, and these effects reverberate through generations. While the ACEs conceptual framework expanded pyramid is a useful model and a visual aid for understanding the historical and present-day dimensions of ACEs in Indigenous communities, a healing conceptual framework is needed to outline a path toward increased community well-being. In this article, we provide a holistic Indigenous Wellness Pyramid that represents the other side of the ACEs pyramid to guide pathways toward healing in Indigenous communities. In this article, the authors describe the Indigenous Wellness Pyramid according to each of the following contrasts with the ACEs pyramid: Historical Trauma—Intergenerational Healing/Indigenous Sovereignty; Social Conditions/Local Context—Thriving Economic and Safe Communities; ACEs—Positive Childhood, Family, and Community Experiences; Disrupted Neurodevelopment—Consistent Corrective Experiences/Cultural Identity Development; Adoption of Health Risk Behaviors—Cultural Values and Coping Skills; Disease Burden and Social Problems—Wellness and Balance; Early Death—Meaningful Life Longevity. We provide examples, supporting research, and implications for implementing the Indigenous Wellness Pyramid.

## 1. Introduction

There are over 450 million Indigenous peoples worldwide who are embedded in diverse cultural and linguistic communities with varied traditions and norms [1]. Historically, in locations globally, Indigenous peoples lived in communities unbounded by international borders, and they established communities in concert with the natural environments and landscapes that they inhabited [2]. Indigenous groups are not monolithic, and there are many unique customs and characteristics of Indigenous communities that are categorized by different names worldwide. While Indigenous cultures vary in terms of language and traditions, there is also similar, shared experience related to the effects of colonization and globalization. Indeed, there is a “unifying thread” among Indigenous people globally that reflects the damaging effects of colonization, and concomitant historical trauma and adverse childhood experiences (ACEs) [3]. Throughout much of the world, Indigenous cultures have experienced changes and colonization that have incurred historic and present-day negative impacts on overall health, as well as social disadvantage and poorer health outcomes compared to the general population [4,5].

These effects of colonization on Indigenous peoples are associated with ACEs, and over two decades of research consistently demonstrates a moderate correlation between ACEs and a wide range of negative health and social outcomes [6,7]. Systems of care that serve Indigenous communities, much like individuals who are under stress and trauma, are often operating in survival mode. This positions the systems of care into a reactive state (rather than a responsive mode), with less of an ability to plan, create, reflect, debrief, and innovate. As a result, systems of care for Indigenous peoples need improved strategies, interventions, or curricula that are going to provide a healing framework for individuals, and community systems that are cumulatively impacted by various types of trauma and adverse events. A systemic, coordinated, and wellness-focused framework for intervention and prevention can help to create momentum and to revitalize healing within Indigenous communities. To address these challenges in Indigenous communities and systems of care, we will present the Indigenous Wellness Pyramid as a wellness-oriented framework and path toward healing from systemic and cumulative community-wide trauma, and to promote wellness mobilization within Indigenous communities.

## 2. The ACEs Pyramid and ACEs

The ACEs pyramid is a model that is used to illustrate and to describe the positive correlation between ACEs and negative health outcomes such as disease and early death [8]. The ACEs pyramid is based on a developmental model that progresses from the bottom of the pyramid, representing conception, to the top of the pyramid, representing death [8]. The eight areas of the ACEs pyramid represent mechanisms by which ACEs can impact health throughout the life span. Those mechanisms, from conception to death, moving from the bottom of the pyramid to the top, are Generational Embodiment/Historical Trauma; Social Conditions/Local Context; Adverse Childhood Experiences; Disrupted Neurodevelopment; Social, Emotional, and Cognitive Impairment; Adoption of Health Risk Behavior; Disease, Disability, and Social Problems; and Early Death [9].

ACEs are disturbing and potentially traumatizing events that occur in childhood, and these fall into broad categories of abuse, neglect, and household dysfunction [10]. Since the first landmark study on ACEs in the 1990s, research findings have consistently demonstrated that, while not causal, increased ACEs that are experienced in childhood are significantly and positively correlated with negative outcomes in adulthood in areas of mental health, physical health, risky behavior, and substance abuse [9,11,12,13]. Given these findings, it is of particular concern for Indigenous communities that ACEs have been found to be higher in Indigenous populations than in non-Indigenous populations [6]. Because of these far-reaching negative health effects of ACEs, it is not surprising that robust research indicates that Indigenous populations commonly have significantly poorer health and social outcomes compared to non-Indigenous populations in domains such as life expectancy, maternal mortality, child malnutrition, child obesity, educational attainment, and suicide rates [6,14].

## 3. The Indigenous Wellness Pyramid

Understanding risk factors and deficit-based explanations of poor health and social outcomes has a long history in psychological and health sciences, and this provides a useful but incomplete framework for conceptualizing problems [15]. An excessive focus on a deficit model undermines clarity about routes toward increased well-being and resilience that have the potential to uplift individuals and communities. Resilience science represents a shift toward a focus on pathways toward resilience and positive adaptation [15]. Indeed, resiliency research points toward the importance of conceptual frameworks that clarify positive pathways toward wellness and health promotion that are scalable across system levels such as individual, family, community, and the broader society [15,16,17]. The Indigenous Wellness Pyramid, as displayed in Figure 1, provides such a framework for Indigenous peoples, communities, and systems of care. This conceptual framework has been developed from the existing work of the primary author at the University of Montana National Native Children’s Trauma Center, and the realization that broader community-wide efforts can facilitate or inhibit interventions to sustainably heal individuals over longer periods of time. This model is a conceptual framework that is yet to be implemented by Indigenous communities. Section 3.1, Section 3.2, Section 3.3, Section 3.4, Section 3.5, Section 3.6 and Section 3.7 of this article contrast domains of the ACEs pyramid with the Indigenous Wellness Pyramid. It is the community grounded cumulation of these strategies into a comprehensive community-wide approach which facilitates systematic and broader infrastructure changes toward healing that bring about stabilization and ongoing healing. This ultimately creates a foundation and conditions for interrupting historical and intergenerational trauma, and preventing adverse childhood experiences.

### 3.1. Historical Trauma—Intergenerational Healing/Indigenous Sovereignty

A foundational feature of a system of healing that is often neglected is the need to address historical and intergenerational trauma. To heal in this area, service systems and providers must heal the historic mistrust with Indigenous communities so that services can be provided from a trusting relationship and a place of mutual understanding. Uplifting Indigenous communities to empower them to decide how healing happens in their communities reestablishes their power and ownership in the development, implementation, and the sustainability of a comprehensive community-wide plan. Empowerment, voice, and choice are trauma-informed principles for developing a trauma-informed approach [18]. This effort involves going beyond a general principle of sovereignty and healing, to actively restore decision-making power to Indigenous communities. Honoring the decision-making power of tribal governments, tribal program leaders, community leaders, elders, and other community stakeholders can restore trust and Indigenous sovereignty through Indigenous-led decision-making.

Who decides how healing happens in an Indigenous community is a critical question to keep at the forefront of efforts toward promoting health and sovereignty in Indigenous communities. It was Gordon Paul [19] who originally posited the famous question, “*What* treatment, by *whom*, is most effective for *this* individual, with *that* specific problem, and under *which* set of circumstances?” This question could be expanded upon when thinking about community-wide healing by asking, *What* healing practices, by *whom*, are most effective for *this* population, with *that* set of problems, and under *which* set of circumstances? Regardless of the particulars, it is emphasized that it is not just *what* systems of care do, but *how* they implement interventions and healing practices that drives therapeutic change. In cultivating healing practices, the process of establishing and engaging in those practices need to be healing as well.

Historical and intergenerational trauma are cumulative and persistent across generations [20]. It has only been in recent decades that it has been more widely recognized and substantiated as a type of trauma [21]. The dominant narrative surrounding Indigenous histories is to “forget about it, move on,” yet the narrative around other historical events are to “never forget.” This effort to disempower and to diminish the narrative contributes to historical unresolved grief, disenfranchised grief, and internalized oppression [22]. The validation, memorializing, repatriation, and reconciliation of these historical events is still work that needs to be done by governments (local, city, county, state, and federal) and the systems of care that work to provide healing. Anyone working with Indigenous communities should have basic competencies about historical and intergenerational trauma broadly, as well as tribal specific histories to contribute to their conceptualization, alliance building, and cultural collaboration [23]. We should switch from asking the question, “What is wrong with this community?” to “What has happened to this community?” [24]. As organizations work to embed trauma-informed care into their policies and practices, they need to address all types of trauma, including systems and community traumatic experiences such as historical and intergenerational trauma, in addition to individual-level trauma.

Indigenous communities have started to make substantial gains toward revitalizing traditional and cultural language, lifeways, healing practices, and moving forward the work of decolonization. It is now well documented that culture is an essential context that underlies prevention and intervention efforts [25]. Although the substantiation of traditional healing practices is controversial to study through western science, Indigenous communities consider them to be practice-based evidence or community-defined evidence because they have been in practice for thousands of years, and this long-standing, historic use demonstrates the evidence of their healing power [26]. Given the diversity that exists between and within Indigenous communities, it is reasonable to support and to provide access to varied methods toward healing (i.e., empirically supported, practice-based, community developed, culturally grounded, etc.) that are decided upon by communities. There are many efforts toward funding that tend to support empirically supported treatments and interventions, with less going toward traditional healing methods [27]. When doing the work of decolonization and honoring tribal sovereignty, systems of care must ground efforts in Indigenous epistemology, honoring both how the community historically and in modern times defines and heals from these issues without outside influence. Then, systems of care can align their services to this effort.

### 3.2. Social Conditions/Local Context—Thriving Economic and Safe Communities

It is difficult to imagine the healing of an Indigenous community amidst ongoing chronic stressors such as ongoing poverty, injustice, unemployment, houselessness, and hunger. Health and well-being are certainly linked, and they have a correlational relationship with the social conditions and resources available. The goal is not only about meeting communities’ basic needs, but also, more importantly, establishing physical and psychological safety. Having more places that are available in Indigenous communities will help to restore a sense of felt safety, justice, and connection to place, will enhance community members’ ability to learn new ways of functioning, and will promote community members engaging intentionally and regularly in healing practices. Focusing on improving social conditions of communities is prevention for community violence, lateral violence, and other types of community traumas. Healing in these ways mobilizes Indigenous systems of care to move beyond just thinking about mental health individually, but also on pathways for planning and action, on a family and community-wide level. One example of developing safe, sustainable, and affordable communities grounded in cultural values and addressing the basic needs of the community is the Eden Model in the Oglala Lakota Tribe in Midwestern United States [28]. In this model, community members have identified seven tenets, based on basic community needs, which provide a foundation for actions aimed at creating a thriving community, such as building an Indigenous Wisdom Center, an Indigenous Generational Housing model, and a passive, regenerative food system [28].

### 3.3. ACEs—Positive Childhood, Family, and Community Experiences

As stated previously, research on ACEs has repeatedly demonstrated a significant correlation between higher ACEs and negative physical health outcomes, social outcomes, and mental health outcomes [9,11,12,13]. While important in conceptualizing the problem, this focus on the negative life trajectory has limitations as a framework for establishing pathways toward increased well-being. In contrast to the deficit model, resiliency science [15,16,17] and PACES (protective and compensatory experiences) [29] have forged positive pathways toward health promotion, resiliency, and the leveraging of protective factors. PACES are individualistic, but it is important that this is expanded upon to be more collectivistic through the inclusion of family and community PACES. It is also noteworthy that resilience science has been demonstrated to be scalable in application to individual and more systemic levels [15,16,17]. As one example, in New Zealand, Māori-specific support for Māori people impacted by earthquakes was shown to enhance the well-being of communities that were impacted [30]. Culturally specific processes, such as support programs grounded in Indigenous values such as kinship, land, and ancestral gathering places, helps promote Indigenous community well-being through culture-centered responses to adversity. Indeed, researchers have found a dose response correlation between PACES and later mental health and social and emotional support in adulthood, regardless of how many ACEs a person experienced [31].

Creating positive childhood, family, and community experiences is not merely the absence of adverse childhood experiences or reducing risk factors, but enhancing protective factors. Examples include:Positive identity development, family and community belonging.Individual pride, family pride, and community pride.Self-actualization, family actualization, and community actualization.

These also can be defined by the community (what do they consider are positive childhood, family, and community experiences?). Enhancing the regularity and intensity with which individuals, families, and communities are immersed in these experiences can promote individual and community well-being.

### 3.4. Disrupted Neurodevelopment and Social/Emotional/Cognitive Impairment—Consistent Corrective Experiences and Cultural Identity Development

Because trauma causes negative automatic associations with various environmental and interpersonal stimuli, it becomes important to provide corrective experiences to similar stimuli to weaken trauma reminders, reduce fear, and provide corrective experiences [32]. In some Indigenous communities, there are various beliefs about healing from trauma; one such belief involves helping the person return to the place of the trauma, if possible, to create a healing experience [33]. Often, this is referred to in vivo exposure and reprocessing [32].

Trauma is such a devastating and severe experience that individuals often must recreate their identities to redefine meaning surrounding the experience itself, but also determine how it will or will not define them into the future [34]. In response to historical and intergenerational trauma, it can mean redefining family and community narratives about the trauma that it has experienced. Indeed, this too has often focused on individual corrective experiences and identity re-creation, but it should also include familial and community corrective experiences and narratives. Particularly in response to historical trauma, Indigenous community members must not only redefine who they are as humans, but also in relation to their ethnic and cultural identities as Indigenous people. This can involve recognizing historical trauma’s impact on identity development, community belongingness, and pride. Family and community cultural roles can then be restored.

### 3.5. Adoption of Health Risk Behaviors—Cultural Values and Coping Skills

The adoption of health risk behaviors is oftentimes an unconscious act [35]. Often, individuals are looking for a way to cope with the traumatic experiences that they have endured, and they seek relief without thinking about the long-term negative health outcomes that are associated with the behaviors. There have been substantial efforts to include social emotional learning in schools to help children to learn earlier how to cope with difficulties, since these are often skills that are not always learned in childhood [36]. The good news is that coping skills can be learned at any age, and they can be learned in coordination with cultural values. When learning self-regulation, drumming and/or singing can be incorporated into self-regulation practices. Emotional psychoeducation can include traditional language when learning about feeling states. Cultural practices such as smudging can be incorporated into learning about relaxation or interpersonal conflicts. Specific knowledge about how community members historically engaged in coping with difficult events can be particularly meaningful.

### 3.6. Disease Burden and Social Problems—Wellness and Balance

Over time, the fields of health and mental health have continued to become more holistic and verifying of the brain–body connection. The cognitive triangle is a well-known model for learning about the connection between emotions, cognitions, and behaviors. Still missing from such models is the focus on spirituality. A recent meta-analysis found that 30% of study participants diagnosed with PTSD experienced a decrease in religious/spiritual beliefs after experiencing a single trauma [37]. This impacts wellness, because for many people, religious/spiritual beliefs are a foundation for meaning in life. While some meta-analytic research indicates that trauma can negatively impact religious/spiritual beliefs [37], a separate meta-analysis examined the impact of spirituality/religiosity, in which researchers found a moderate and positive correlation (r = 0.40) between spirituality/religiosity and resilience [38]. The bi-directional relationship between trauma and spirituality underscores the importance of spirituality as a dimension of wellness frameworks in Indigenous communities. Spirituality can be addressed while honoring diverse religious beliefs that individuals may hold. Ignoring this important aspect of well-being altogether is a detriment to the individual and the community in their healing journey.

Even more important than a holistic worldview of well-being is honoring Indigenous communities’ understanding of wellness and balance uniquely as different groups. These perspectives are often in contrast to Western epistemologies. For example, in Indigenous communities, wellness is not defined by the absence of disease, pain, trauma, loss, or negative feelings, but experiences that sometimes unfortunately occur, and an effort to restore balance and to honor healing processes over time [39]. This Indigenous epistemology honors the fact that unfortunately, diseases, pain, trauma, loss, or negative experiences are a part of living, and so, holistic healing encompassing the mind, body, emotion, and spirit are needed to restore and to maintain balance in one’s life. This also honors that healing occurs over time and that it can be interrupted by external factors. In contrast, mainstream society focuses on promoting the best evidence-based interventions to provide instant symptom reduction. This focus may not align with how the community defines wellness and balance overall or be in alignment with how success or progress toward healing is determined.

### 3.7. Early Death—Meaningful Life Longevity

It has been well substantiated that there is a lower life expectancy of Indigenous people, in contrast to the general population [40]. All of these healing efforts would eventually restore meaningful life longevity. However, this is not necessarily the goal, in and of itself. The overall goal is more meaningful. It is to have elders in Indigenous communities for longer, to promote future intergenerational healing, and to forge pathways toward increased sovereignty and community wellness. In essence, Indigenous communities can achieve self-actualization, family actualization, and community actualization, honoring how the community would define it.

## 4. Conclusions

This healing framework encompasses flexibility and honors decision-making authority in deciding how healing happens. Healing is more than just one provider and one client. It takes individuals, programs, systems, and entire communities mobilizing to create healing change; moving beyond crisis mode to plan for future generations, recognizing this is going to be a long-term process, and validating the fact that it is difficult to heal in the face of enduring stress, trauma, grief, and loss. This framework restores the hope for community-wide mobilization and healing, while honoring the work that has already occurred to provide healing in Indigenous communities.

## Figures and Tables

**Figure 1 ijerph-20-04108-f001:**
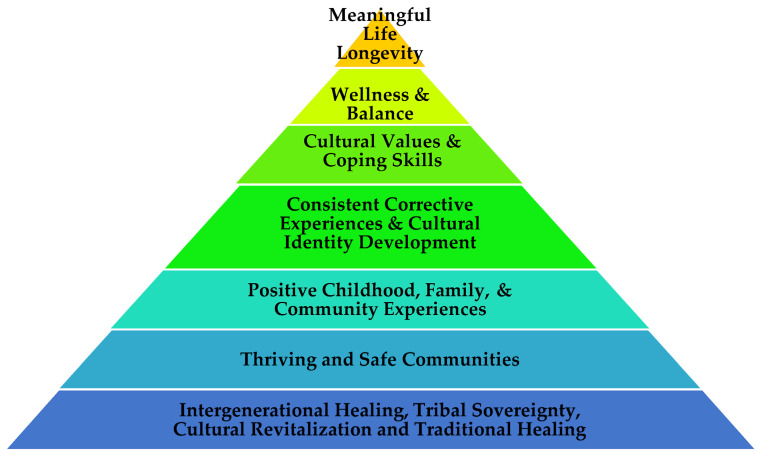
The Indigenous Wellness Pyramid provides a scalable, positive pathway toward increased well-being in Indigenous communities, and it is the opposite side of the ACEs pyramid. The shift toward resiliency, positive pathways, health promotion, and protective factors is reflected in the seven areas of the model.

## Data Availability

Not applicable.

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
