# Peer review of "The Other Side of the ACEs Pyramid: A Healing Framework for Indigenous Communities"

_ijerph, 2023, doi:10.3390/ijerph20054108_

Round 1

Reviewer 1 Report

Thank you for the opportunity to review this important manuscript. In this clearly written manuscript, the authors present important background/foundational information describing current knowledge about the intersectionality of trauma impacts on the wellness of Indigenous people. The focus of the paper describes their conceptual Indigenous Wellness Pyramid framework model of wellness and healing from historical and ACE-related trauma within Indigenous populations/communities. This is a critically important contribution and foundation for future empirical testing and cultural vetting. Thank you for your work.

Author Response

• No revisions needed as reviewer comments were complimentary

Reviewer 2 Report

Please see the attached

Author Response

Reviewer 2
Many but not all individuals who experience ACEs develop problems raising critical questions about what is “cause” and what is “effect.”
• Lines 44, 60, 74-75, we emphasize correlation and positive relationship to address the reviewers perspective about causality or determinism
…the proposed solution reduces a complex set of circumstances to, ostensibly, a situation before colonization to bring about healing.
• Lines 99-105 were written to discuss that that tribal sovereignty alone or any strategy at each level alone is an incomplete conceptualization of the framework and stating the intention more clearly
However, this issue requires attention in the manuscript rather than implying that individual indigenous communities are monolithic in voice and worldview. Indeed, to operationalize the model, the authors need to offer guidance (other than general examples) on realities of social life, like power dynamics within the community, the right balance of tradition and change, and normative expectations for behavior. Related concerns arise from the appearance that culture and its artifacts (e.g., epistemology, everyday life practices) are fixed or static.
• Lines 33, 35-36, 272-273 address that we are not implying that individual Indigenous communities are monolithic in voice and worldview
What, for example, does cultural restoration mean?
• The authors did not use this term, so we agreed we did not need to define it

Reviewer 3 Report

It is interesting to read the paper on a pyramidal healing framework for indigenous communities grounding in Indigenous epistemology. The paper is built on decades of research on the well-being of indigenous communities around the world. To make each case of the indigenous wellness pyramid more clearer or stronger, for instance the section on "Disease Burden and Social Problems – Wellness and Balance", the authors can give appropriate example through the description of one such practices of indigenous community on the "effort to restore balance and honor healing processes overtime". This will particularly make non-indigenous readers understand clearer.

Author Response

Reviewer 3
…the authors can give appropriate example through the description of one such practices of indigenous community on the "effort to restore balance and honor healing processes overtime"
• Lines 276-284 were added to provide further explanation

Reviewer 4 Report

I very much like this paper and enjoyed reading it. I have some very minor suggestions.

In the introduction, it doesn’t seem necessary to idealize previous Indigenous culture to make the point. Whether or not there is one world-wide culture that was unbounded by international borders and lived in concert with natural environments does not change the overall argument, and I think it will make some people worry that it’s too sweeping and not scientific. The shared experience of colonization and globalization and that “unifying thread” seems like the place to really concentrate the rhetoric of this article. That the base of the pyramid then very appropriately recaptures a sense of tradition can be based in cultural revitalization and not in contested arguments about the history of particular Indigenous peoples.

Do you really need “evidence-based strategies” (l.48)? Can’t they just be well thought out and concretely situated? I personally think there’s good reason to think “evidence-based” is part of the globalization paradigm because it seeks to centralize and depersonalize judgment about what should be done. The discussion of evidence in ll. 144-46 would also be more consistent without making an empty nod toward evidence-based practice here.

In the pyramid, why not call “Life Longevity,” “Long Life” (or, in light of l.266 ff., “Meaningful Longevity”)?

I think the 9/11 example is going to be misread by your audience, and I think it’s a bit unclear. Are you saying that it was a good healing practice for the larger U.S. “community” to say “never forget” and that Indigenous peoples with similar traumas should also have access to that healing practice? Or are you saying that there’s so much hypocrisy in the “never forget” that it was just empty for them to say it, and it didn’t provide real healing in any case? If there’s no comparison between a single unexpected attack and a brutal history of exploitation, that’s a different argument than saying that you guys got to treat 9/11 as traumatic, please recognize that we also deserve healing around our trauma – I think you can clarify (or pick a different example) in a way that can pick out either one side or the other of the point. More broadly, your strongest argument is that systems of care should align their services to the efforts of decolonization and honoring tribal sovereignty and Indigenous epistemology – it’s undercut if you compare 9/11 (where the attackers saw themselves as anti-colonialist, but who weren’t making a claim to doing decolonizing work in a grounded fashion) with the original colonizing violence. I think a better example, or better working out of that example, would be helpful.

Again, those are small points. Overall, I really enjoyed it.

Author Response

Do you really need “evidence-based strategies” (l.48)? Can’t they just be well thought out and concretely situated? I personally think there’s good reason to think “evidence-based” is part of the globalization paradigm because it seeks to centralize and depersonalize judgment about what should be done. The discussion of evidence in ll. 144-46 would also be more consistent without making an empty nod toward evidence-based practice here.
• Deleted references to evidence-based and kept strategies as we agreed with the comment

In the pyramid, why not call “Life Longevity,” “Long Life” (or, in light of l.266 ff., “Meaningful Longevity”)?
• Changed the label to “Meaningful Life Longevity”

I think the 9/11 example is going to be misread by your audience, and I think it’s a bit unclear.
• Deleted reference to 9/11 specifically and made this a broader example